# Prevalence of Vitamin D and Calcium Deficiency and Insufficiency in Women of Childbearing Age and Associated Risk Factors: A Systematic Review and Meta-Analysis

**DOI:** 10.3390/nu14204351

**Published:** 2022-10-17

**Authors:** Erika Aparecida da Silveira, Letícia de Almeida Nogueira e Moura, Maria Clara Rezende Castro, Gilberto Kac, Maria Claret Costa Monteiro Hadler, Priscilla Rayanne E. Silva Noll, Matias Noll, Andréa Toledo de Oliveira Rezende, Felipe Mendes Delpino, Cesar de Oliveira

**Affiliations:** 1Health Science Graduate Program, Faculty of Medicine, Federal University of Goiás, Goiânia 74605-050, GO, Brazil; 2Department of Epidemiology and Public Health, University College London, London WC1E 6BT, UK; 3Nutritional Epidemiology Observatory, Department of Social and Applied Nutrition, Institute of Nutrition Josué de Castro, Federal University of Rio de Janeiro, Rio de Janeiro 21941-902, RJ, Brazil; 4Graduate Program in Nutrition and Health, Faculty of Nutrition, Federal University of Goiás, Goiânia 74605-050, GO, Brazil; 5Department of Obstetrics and Gynaecology, University of São Paulo, São Paulo 05403-000, SP, Brazil; 6Campus Ceres, Federal Institute Goiano, Ceres 76300-000, GO, Brazil; 7Postgraduate Program in Nursing, Federal University of Pelotas, Pelotas 96010-610, RS, Brazil

**Keywords:** pregnancy, micronutrients, vitamin D deficiency, hypocalcemia, diabetes, obesity

## Abstract

Vitamin D deficiency and insufficiency as well as low serum calcium levels can trigger negative health outcomes in women of childbearing age. Therefore, we aimed to estimate the prevalence of serum vitamin D and calcium deficiencies and insufficiencies and associated risk factors in Brazilian women of childbearing age and to assess whether there are differences in prevalence according to regions of the country and the presence or absence of pregnancy. The systematic literature review was performed using the following databases: PubMed, LILACS, Embase, Scopus, and Web of Science. Cross-sectional, cohort, and intervention studies were included. Among pregnant women, the prevalence of vitamin D deficiency ranged from 0% to 27% and of vitamin D insufficiency from 33.9% to 70.4%. Among non-pregnant women, the prevalence of vitamin D deficiency ranged from 0% to 41.7% and of vitamin D insufficiency from 38.5% to 69.3%. We found a high prevalence of vitamin D deficiency and insufficiency in women of childbearing age, with insufficiency affecting more than half of these women. The highest prevalence of vitamin D deficiency and insufficiency was observed in the South region. It was not possible to assess the prevalence and factors associated with calcium deficiency.

## 1. Introduction

Nutritional deficiencies affect individuals worldwide [1], especially women of childbearing age during pregnancy [2]. During this period, a woman’s body faces major physical and physiological changes due to the needs of the growing baby [3], making them vulnerable to vitamin and mineral deficiencies [4].

In an observational study conducted in Southern Korea, 89% of women of childbearing age had low concentrations of vitamin D [5], and another observational study of pregnant Iranian women showed a 58% prevalence of vitamin D deficiency during pregnancy [6]. Although there is no agreement on the optimal range of vitamin D deficiency, it is predominantly characterized by serum 25(OH)D concentrations below 25–30 nmol /L (10–12 ng/mL) [7]. Vitamin D deficiency is highly prevalent even in countries with high solar incidence [8] and is a serious public health problem. Therefore, it is important to understand the profile of this deficiency in women of childbearing age.

The evidence showing that vitamin D deficiency can trigger negative outcomes for both the mother and child, including hypertension in women of childbearing age [9], gestational diabetes [10] and negative adverse perinatal outcomes [11] such as first trimester miscarriages [12] has resulted in a growing public health concern at primary care level and the need of public policies at the national level. Vitamin D deficiency has also been associated with some extra-skeletal damages, such as osteoporosis, chronic musculoskeletal pain, muscle weakness, and an increased risk of falling [13,14,15,16]. The negative adverse perinatal outcomes [17] include impairment of anthropometric measurements of the neonate such as birth weight, length, and head circumference [18] in addition to a higher risk of premature rupture of membranes [10] and miscarriage [19]. At birth, it can be related to infantile eczema, nutritional rickets, severe hypocalcemia, and other orthopedic complications in neonates and children [20].Therefore, vitamin D deficiency in combination with periodontal disease has been associated with preterm and low birth weight [21].

Vitamin D is a hormone that can regulate calcium metabolism [22]. Calcium is part of the mineral component of bone and plays a role in nerve and muscle function, intracellular signaling, electrophysiology of the heart, and coagulation [23]. Therefore, a stable concentration of calcium is important to maintain homeostasis of the body systems [24]. Hypocalcemia develops when the calcium concentration is below 3 mg/dL (0.75 mmol/L) [25]. An observational study conducted in Latin America found that a high number of women of childbearing age have an inadequate calcium intake, especially in Peru (97.74%), Costa Rica (95.88%), Brazil (95.16%), and Chile (93.23%) [26]. Calcium deficiencies trigger disorders ranging from mild (such as perioral paresthesia) to severe (such as cardiac arrhythmias) [27]. During pregnancy and lactation, the interaction of vitamin D with calcium is important for preventing neonatal rickets and reducing the risk of preeclampsia [28], gestational diabetes, and premature childbirth [29].

Despite the importance of this topic, there is only one systematic review conducted in Brazil on the prevalence of vitamin D deficiency and insufficiency [30]. Our review differs from the previous study, as it included rigorous data from well-conducted studies stratified by gestation status and presented regional differences by the five great geographic Brazilian regions. In addition, our review is the first to estimate calcium deficiency and insufficiency and their associated risk factors in Brazilian women of childbearing age. Therefore, this study aimed at estimating the prevalence of serum vitamin D and calcium deficiency and insufficiency and their associated risk factors in Brazilian women of childbearing age and assessing whether the prevalence differs according to the presence or absence of pregnancy and region of the country. The results of this review contribute to the development and improvement of public health policies aimed at women of childbearing age, guiding the creation of intervention protocols for reducing deficiencies in vitamin D and calcium, which are essential micronutrients for the health of this population.

## 2. Materials and Methods

This systematic review and meta-analysis were conducted in accordance with the preferred reporting items for systematic reviews and meta-analysis (PRISMA) protocols. This review follows the PICO structure, with “P” (population) being women of childbearing age, “I” being no intervention, “C” being comparison between pregnant and non-pregnant subgroups and region, and “O” (outcome) being the prevalence calcium and vitamin D deficiency. The protocol (CRD42020207850) of this systematic review was registered on the PROSPERO platform and published under the title *Prevalence of vitamin D and calcium deficiencies and their health impacts on women of childbearing age: a protocol for systematic review and meta-analysis* [31].

### 2.1. Search Strategy and Databases

The systematic literature review was performed using the following databases: PubMed (National Library of Medicine), LILACS (Latin American and Caribbean Health Sciences Literature), Embase, Scopus, and Web of Science. The search was conducted in December 2021 to identify articles published until that date for inclusion in this review. The review was updated in March 2022. The search strategy included relevant keywords related to vitamin D deficiency and specific terms related to calcium deficiency to ensure that all articles of interest were identified. Details describing the entire methodology and search strategy with the terms are published in the protocol article [31].

### 2.2. Eligibility Criteria

#### 2.2.1. Inclusion Criteria

Studies that provide data on the prevalence of serum calcium and/or vitamin D deficiency in women of childbearing age (15–49 years or menarche and menopause).Studies with representative population-based samples in hospitals, health centers, or outpatient clinics.Prevalence data in women of different age groups, such as adolescents, pregnant women, lactating women, and premenopausal adult women.Studies with a cross-sectional design and data from longitudinal studies (cohort studies) or intervention studies, such as clinical trials or community trials, provided they had prevalence information for a specific time. Articles in English, Portuguese, and Spanish were included.

#### 2.2.2. Exclusion Criteria

○Opinion articles, comments, or editorials.○Duplicate articles, i.e., the same study found in different databases.○Articles with the same database/population/sample, in which case the study with the largest sample size was considered.○Articles with primary data not accessible even after request to the authors.○Case-control articles, narrative reviews, and case series.○Studies conducted among female athletes of any sport.○Studies conducted among women with the following specific diseases: autoimmune diseases such as lupus, psoriasis, thyroiditis, rheumatoid arthritis, and multiple sclerosis; eating disorders such as anorexia and bulimia; hematological diseases such as thalassemia and sickle cell disease; respiratory diseases such as chronic obstructive pulmonary disease, asthma, pneumonia, respiratory infections, and tuberculosis; chronic diseases such as heart failure, kidney failure, liver disease, chronic kidney disease, heart disease, nephrotic syndrome, AIDS, inflammatory bowel disease, hypo- or hyperthyroidism, sepsis, and cancer; genetic diseases and syndromes such as vitamin D receptor mutation, cystic fibrosis, and Prader–Willi syndrome; neurological or psychiatric disorders such as epilepsy (or antiepileptic medication use), attention deficit hyperactivity disorder, and schizophrenia.○Studies conducted among post-surgical patients, patients with trauma or burns, or patients undergoing recent treatment for fractures or orthopedic/osteoarticular diseases.○Studies conducted among patients undergoing intensive, urgency or emergency, or palliative care.○Studies with fewer than 50 participants.○Studies conducted among indigenous women.

### 2.3. Reviewer Training

The authors responsible for assessing the eligibility criteria of the articles were trained. An eligibility test was performed with 50 titles and abstracts before coding the articles. The reviewers also received training in the instruments used to assess the risk of bias through five articles that were not included in the review. Rayyan and Mendeley software were used for the selection steps, the first being the selection of studies and the second being the exclusion of duplicate articles.

### 2.4. Review Process

After completing the search strategy, the identified articles were gathered and imported into the Mendeley software. Duplicate articles were excluded. The articles were selected by two independent reviewers (L.A.N.M. and M.C.R.C.). The titles were read first and then the abstracts. Finally, the entire article was read. Disagreements between the two reviewers were resolved by a third reviewer (E.A.S.). Eligibility was determined according to inclusion and exclusion criteria.

### 2.5. Data Extraction and Risk of Bias Assessment

Data were extracted using a table considering the following aspects: author/year, type of study, study location, age group, sample type/sample size, place of residence (urban/rural), gestational status (yes/no), lactating (yes/no), micronutrients analyzed, technique used, cut-off points and results—that is, prevalence/impact of calcium or vitamin D deficiency. The values of vitamin D are expressed as nanogram per milliliter (ng/mL) or nanol per liter (nmol/L). For conversion purposes, it is enough to multiply ng/mL by 2.5 to obtain the value in nmol/L [32].

The Downs and Black scale used to assess the risk of bias is an instrument with 27 items, but with only 16 items being applicable to observational studies (items 1–3, 5–7, 9–12, 17, 18, 20, 21, 25, and 26). The score was applied to each article according to the number of items, considering the total percentage (0 to 17 points). Low risk of bias was defined as a total score > 70%.

The Grading of Recommendations, Assessment, Development, and Evaluation (GRADE) approach was used to assess the quality of evidence of selected studies. For each study, quality was assigned one of the following four grades: high quality (four filled circles), moderate quality (three filled circles), low quality (two filled circles), or very low quality (one filled circle).

Data were extracted and evaluated by two independent reviewers (L.A.N.M. and M.C.R.C.). Disagreements were resolved by a third reviewer (E.A.S.). A researcher (M.C.R.C.) contacted the authors of the articles to obtain relevant data not reported in their article. Potential conflicts of interest and ethical information from the studies included in the review were also reported.

### 2.6. Statistical Analysis

A meta-analysis was performed to assess the prevalence of vitamin D deficiency and insufficiency in Brazilian women. The analyses were stratified by region of the country and pregnancy status (pregnant or non-pregnant). Given the high heterogeneity between studies, random-effects models were used to reduce differences. We calculated the mean of the vitamin D deficiency and/or prevalence values stratified by gestational trimester in the studies and pooled them for meta-analysis. The analyses were performed using the R language (version 4.1.0) and the Meta package (version 6.0-0) and metaprop command. When the analyses included more than ten publications, a funnel plot was used to assess the asymmetry between studies.

## 3. Results

Of the 149 articles with data on serum calcium and vitamin D deficiency and insufficiency in Brazilian women of childbearing age initially identified in the databases, 143 articles remained after the exclusion of duplicates. After applying the eligibility criteria, 73 articles were selected for full reading, of which 16 were included in the systematic review. Seven articles provided information on pregnant women of which five provided information on serum vitamin D and two on calcium (Figure 1).

### 3.1. Prevalence of Vitamin D Deficiency and Insufficiency

The number of women in studies that evaluated vitamin D in pregnant women ranged from 174 to 487, and they were aged between 20 and 40 years, with only two studies including adolescents (Appendix A).

The number of non-pregnant women ranged from 15 to 369, with eight studies having a sample of fewer than 100. Five studies included only adolescents, and seven included adult women or adults and adolescents. The number of women in the two studies that evaluated calcium ranged from 99 to 226, and the age range varied between 16 and 44 years of age (Appendix A).

Most studies were conducted in the Southeast (six articles) and South (four articles) regions. Only two studies were conducted in the Midwest region, and none were conducted in the North region. The cut-off points for both pregnant and non-pregnant women varied widely in the assessment of vitamin D deficiency and insufficiency, with the lowest value for deficiency being ≤10 ng/mL and the highest being <20 ng/mL.

Among pregnant women, the prevalence of vitamin D deficiency ranged from 0% to 27% and the prevalence of insufficiency ranged from 33.9% to 70.4%. One study considered the prevalence of both deficiency and insufficiency in pregnant women, which reached 82.9%. Among non-pregnant women, the prevalence of deficiency ranged from 0% to 41.7% and the prevalence of insufficiency ranged from 38.52% to 69.3%. Two articles reported the prevalence of both insufficiency and deficiency in non-pregnant women, which reached 74.1%.

### 3.2. Factors Associated with Vitamin D Deficiency and Insufficiency

The factors associated with the risk of vitamin D deficiency in pregnant women were being married, the use of vehicles as a means of transportation, blood collection in winter, only face and hands being exposed to the sun, preeclampsia, first pregnancy, adolescence, and low income [33,34]. However, a study with 226 women found no association between vitamin D deficiency and obstetric, biological, and socioeconomic variables [35]. Three studies with non-pregnant women did not investigate associated risk factors [36,37,38]. Five studies [38,39,40,41,42] did not find an association, whereas those that found an association identified the following risk factors: non-white skin, diabetes, serum glucose, homeostatic model assessment for insulin resistance, obesity, and serum calcium. Only two studies evaluated serum calcium, but they did not investigate the associated risk factors [35,39] (Table 1).

### 3.3. Prevalence of Calcium Deficiency and Associated Risk Factors

The hypocalcemia cut-off values ranged from <8.6 to <8.8 mg/dL and only one article with pregnant women showed a 15% prevalence of deficiency (Appendix A).

### 3.4. Quality Analysis of the Evidence

The Downs and Black scale scores ranged from 54% to 100% (Table 2). Fourteen studies scored above 70%, indicating a low risk of bias. The GRADE score, which evaluated the methodological quality, revealed 12 studies with moderate quality, three studies with low quality, and one study with very low quality. All studies declared no conflict of interest and 14 (88%) declared an ethical approval.

### 3.5. Meta-Analysis of the Prevalence of Vitamin D Deficiency and Insufficiency

The prevalence of vitamin D deficiency (Figure 2) in pregnant women was 30% (95% CI: 0.09–0.64; *I*^2^: 99%) and 29% in non-pregnant women (95% CI: 0.19–0.43; *I*^2^: 94%), while the overall prevalence was 29% (95% CI: 0.18–0.42; *I*^2^: 97%). The analysis of deficiency by region (Figure 3) showed a higher prevalence of deficiency in the South region, although without significant differences between regions.

The vitamin D insufficiency (Figure 4) was of 59% (95% CI: 0.51–0.67; *I*^2^: 93%), with no differences between pregnant and non-pregnant women. A higher prevalence of vitamin D insufficiency was observed in the South region than in the other regions, although the confidence intervals did not show significant differences (Figure 5).

A slight asymmetry to the left is observed in the funnel plot for vitamin D deficiency in women of childbearing age, indicating a heterogeneity between studies (Figure 6). The funnel plot for vitamin D insufficiency in women of childbearing age did not show asymmetry, indicating less heterogeneity than the vitamin D deficiency plot (Figure 7).

## 4. Discussion

To the best of our knowledge, this is the first systematic review and meta-analysis to estimate the prevalence of serum concentration of vitamin D and calcium deficiencies and insufficiencies in Brazilian women of childbearing age. The study included 1276 pregnant women and 1436 non-pregnant women and was stratified by Brazilian geographic regions, excluding the Northeast since no study was conducted in this region. We observed a high prevalence of vitamin D deficiency in both pregnant (29.5%) and non-pregnant (29.4%) women without significant differences. The prevalence of insufficiency was 59% and it was similar between pregnant and non-pregnant women. It was not possible to perform a meta-analysis of the studies that estimated the prevalence of calcium deficiency due to the small number of studies, i.e., only two articles. Similarly, we could not conduct a meta-analysis to identify the risk factors associated with vitamin D and calcium deficiency and insufficiency due to the heterogeneity of the investigated factors and the reduced number of studies.

The overall prevalence of vitamin D deficiency (29%) and insufficiency (59%) identified in this meta-analysis were slightly higher than those found in an observational study conducted in Colombia that found a 24% prevalence of vitamin D deficiency and a 47% prevalence of vitamin D insufficiency [50]. An observational study conducted in Pakistan reported that an inadequate vitamin D status is common among women of childbearing age [51]. Another observational study found that 56% of Egyptian women of childbearing age had a vitamin D deficiency or insufficiency [52]. Looking at results from countries other than those with an income level similar to Brazil, we found an observational study conducted in Sweden reporting that more than a third of pregnant women had low levels of 25-hydroxyvitamin D [53]. The high prevalence of inadequate vitamin D status in women of childbearing age can be explained by the fact that a large part of this population lives in urban areas, which is considered a risk factor for vitamin D deficiency. Vitamin D is obtained and synthesized in the skin during exposure to ultraviolet B (UV-B) sunlight (270–300 nm) [54]. Spending time predominantly indoors owing to economic or occupational factors and wearing tight clothing reduces the exposure to sunlight [55]. Urban women generally commute to work, schools, and children’s activities using forms of transport that avoid sun exposure, and the following factors influence the vitamin D levels: day-to-day activities, physical locations where they are performed, location, and geoclimatic conditions [56].

Another influencing factor is the dietary intake of vitamin D, whose levels show a statistically positive association with the Mediterranean diet, regardless of body mass index [57]. The average consumption of this nutrient in adult Brazilian individuals ranged from 2.4 to 4.67 µg [34,45,48], which is below the recommendation of estimated average requirement of 10 µg/day for women of childbearing age (15–49 years), and also according to dietary reference intakes (DRIs) of 15 µg/day for women [58]. An article that evaluated Latin American women pointed to an average vitamin D intake of 1.9 µg by Brazilian women [59], and another article that evaluated dietary intake in Brazilian adolescents (15–19 years old) observed a median vitamin D intake of 1.48 µg/day [44]. Even so, dietary intake was not associated with vitamin D deficiency or insufficiency [34].

The meta-analysis by Brazilian geographic regions showed that the prevalence of vitamin D deficiency and insufficiency was higher in the South region. A possible explanation for this is that the South region has a subtropical climate and, due to its proximity to the Tropic of Capricorn, it is characterized by the lowest temperatures in the country. The main source of vitamin D in humans is UVB radiation (290–315 nm) from the sun on the skin [60], and exposure to UV rays appears to be determinant in the epidermal synthesis of vitamin D [61]. The height of the sun determines the path of rays through the ozone layer, and the intensity of the rays depends mainly on latitude (geographical location), season, and time of day [62]. An observational study conducted in nine European countries showed that the availability of UVB radiation decreased with increasing latitude, and the availability of UVB in the winter months was too low to allow for the cutaneous synthesis of vitamin D [63]. The mean dietary calcium intake is 625.1 to 738.72 mg/day [37,45,48]. According to Herrera-Cuenca (2021), 95.16% of Brazilian women had an inadequate calcium intake, which was also below the recommendation (1000–1200 mg/day). 

It is important to highlight that we did not find studies conducted in the North region of Brazil, which is characterized by pockets of poverty and the worst indicators of infant mortality and health in the country. Only one study in the Midwest region and three in the North region described regional inequalities, not only from the socioeconomic point of view but also from the perspective of promotion of scientific research.

Vitamin D deficiency cut-offs varied widely in the studies included in this review. This is in line with a systematic review including 33 observational studies that showed wide variations in the cut-off values used by studies to determine vitamin D deficiency [30]. The public health vitamin D food fortification or supplementation programs in Brazil are not mandatory [34,46], which is unusual among women receiving prenatal care from public service [34] and might be challenging due to the differing recommendations of vitamin D status, based on studies conducted in high latitude countries with older Caucasian populations [48]. Therefore, further studies are needed to deepen the discussion of optimal cut-offs for specific vitamin D levels for the Brazilian population [48].

The risk factors most associated with vitamin D deficiency in pregnant women in the studies included were being married, the use of vehicles, sun exposure of only the face and hands, preeclampsia, and low income. Being married was identified as a risk factor for vitamin D deficiency in an observational study in Saudi Arabia [64], whereas low socioeconomic status was associated with vitamin D deficiency in an observational study in China [65]. Furthermore, a systematic review of 13 observational studies demonstrated that vitamin D deficiency was associated with an increased risk of preeclampsia in pregnant women [66]. Vitamin D deficiency has also been associated with obesity since people with obesity may be more sedentary, perform less outdoor activities, and therefore be less exposed to sunlight [67]. In addition, adipose tissue is responsible for vitamin D sequestration, resulting in volumetric dilution of ingested or cutaneous synthesized vitamin D3 [68]. A prospective cohort study conducted in Italy showed that men have higher vitamin D concentrations than women across all body mass index classes [69]. Furthermore, women with a vitamin D deficiency had a higher percentage of fat mass when compared to men with a vitamin D deficiency, which can be explained by the fact that women have more localized fat than men [70]. Although several studies have addressed some factors associated with vitamin D deficiency, there is no consensus about these associations. A cross-sectional study conducted in Brazil demonstrated the low prevalence of vitamin D deficiency in severe obesity (10%) and also identified that serum and dietary vitamin D were not associated with metabolic syndrome [71]. Therefore, more research is needed on these factors to guide public health policies in the development of action plans to reduce the rate of vitamin D deficiency and consequently avoid its negative outcomes in women of childbearing age.

A possible limitation of this study was that most of the articles showing vitamin D deficiency and insufficiency were conducted in the Southeast and South regions of the country (63%), which can be considered a bias with respect to obtaining an overview of the country. In most of the articles included in this systematic review, women who supplemented vitamin D were excluded from the study samples [34,37,38,40,41,43,45,46,47,48]. As well as not evaluating the dietary intake of vitamin D, not taking into consideration the use of supplements may be a bias in the study, as it may interfere with vitamin D metabolism [72,73]. Furthermore, it was not possible to perform a meta-analysis of calcium deficiency and the factors associated with vitamin D and calcium deficiency in women of childbearing age due to the low number of studies found. Therefore, the findings should be interpreted with caution. The same limitation occurred for the identification of associated risk factors. However, the strengths of this systematic review include the use of scales that assessed the methodological quality of the studies included as well as the absence of restriction on the year of publication. Another positive aspect relates to the fact that the selection and inclusion of articles were conducted separately by two researchers and the disagreements were resolved by a third reviewer to ensure consistency and rigor in the application of the eligibility criteria. In addition, although another systematic review estimated the prevalence and insufficiency of vitamin D, our review was the only one able to identify the associated factors, as well as to assess the risk of bias of the included studies.

This study highlights the high prevalence of vitamin D insufficiency in both pregnant and non-pregnant women. Thus, we highlight the need for the development of public policies, with a focus on preventing and minimizing this problem. From the point of view of clinical implication, national health education campaigns, focusing on behavioral measures such as 10 minutes of daily sun exposure and a balanced diet [74,75] as well as outdoor physical activity practices [23], may be sufficient to expand women’s knowledge, favoring the achievement of adequate serum levels of vitamin D and calcium.

The education on and promotion of health, including dietary intake that meets nutrient and micronutrient needs, with the use of supplementation and food fortification if necessary, may constitute policies to improve nutritional deficiencies in general [26,58,76]. However, the gaps in knowledge about calcium and vitamin D deficiency and associated factors in Brazilian women are diverse, making it impossible to establish public policies due to insufficient evidence, particularly for different ethnic/racial groups [34,48]. In this sense, the results of this study may contribute to fostering policies to be increased, aiming at the development of research on calcium and vitamin D deficiency in Brazilian women of childbearing age, as well as at the identification of potential risk and protection factors. We emphasize that no research was conducted in the North region and only two studies were conducted in the Center-West region. Therefore, the development of research on serum calcium deficiency in all Brazilian regions and vitamin D deficiency in the North, Northeast, and Midwest regions is recommended to enrich the discussion related to actions and clinical implications. We also suggest prioritizing the regions with the most gaps in the country’s research funding notices.

According to the Brazilian Society of Endocrinology and Metabology, the diagnosis of serum vitamin D deficiency should be performed in pregnant and lactating women [24]. Considering that the prevalence of vitamin D deficiency, in this systematic review, did not differ between pregnant and non-pregnant women, it is necessary that women of reproductive age are also tested to monitor their nutritional status of vitamin D.

## 5. Conclusions

This systematic review and meta-analysis showed a high prevalence of vitamin D deficiency and a higher prevalence of vitamin D insufficiency in Brazilian women of childbearing age. There was no significant difference between pregnant and non-pregnant women. We found a higher prevalence of vitamin D deficiency and insufficiency in the South region. It was not possible to carry out a meta-analysis of calcium deficiency, as only two articles addressed the topic, with one presenting only its prevalence and the other just the mean serum calcium levels.

This systematic review revealed a wide variability in the cut-off points used to classify vitamin D deficiency and insufficiency. Therefore, the results demonstrate the importance of standardizing cut-off points to facilitate comparisons between studies and to obtain accurate and reproducible information about the prevalence of vitamin D deficiency and insufficiency. Considering that vitamin D deficiency and insufficiency was common in women of childbearing age in Brazil, public health policies aimed at primary healthcare should be conducted with a focus on preventing vitamin D insufficiency in this population.

## Figures and Tables

**Figure 1 nutrients-14-04351-f001:**
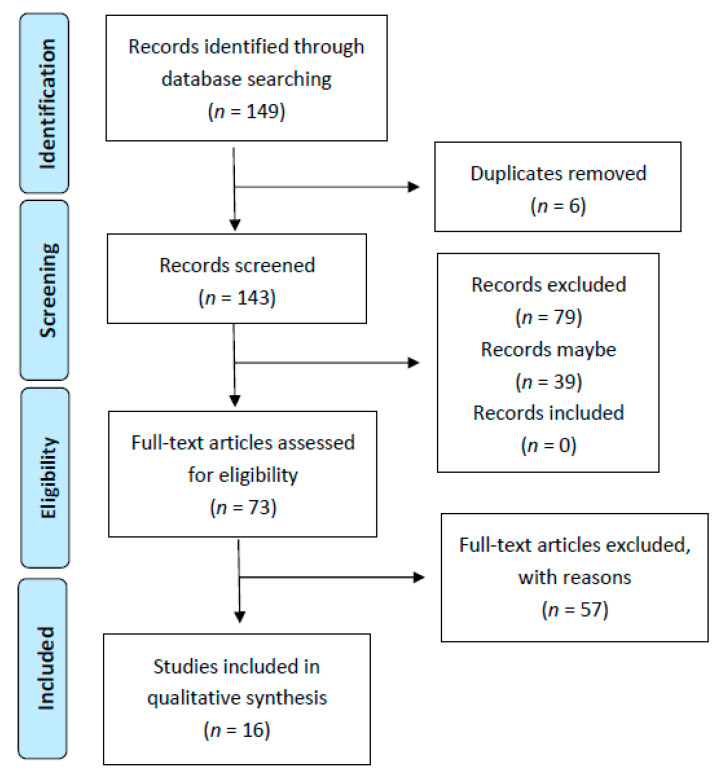
Flowchart of the search process.

**Figure 2 nutrients-14-04351-f002:**
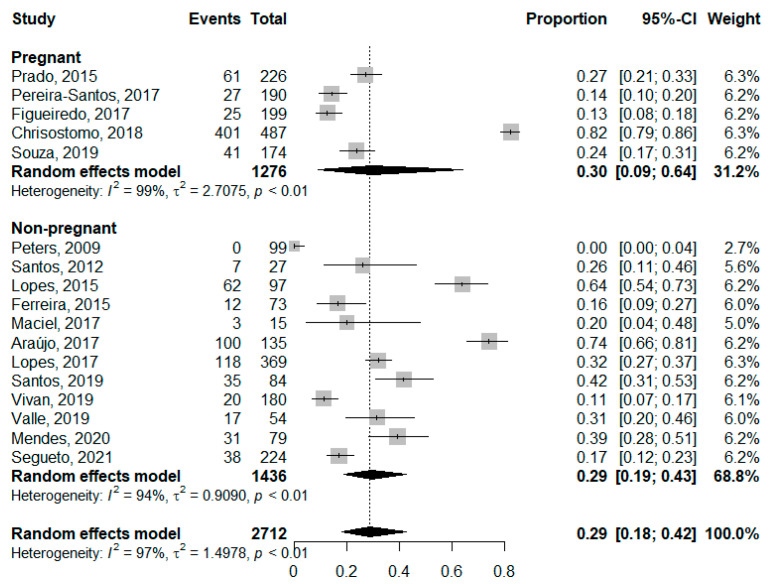
Prevalence of Vitamin D deficiency in childbearing women stratified by pregnancy status. [33,34,35,36,37,39,40,41,42,43,44,45,46,47,48,49].

**Figure 3 nutrients-14-04351-f003:**
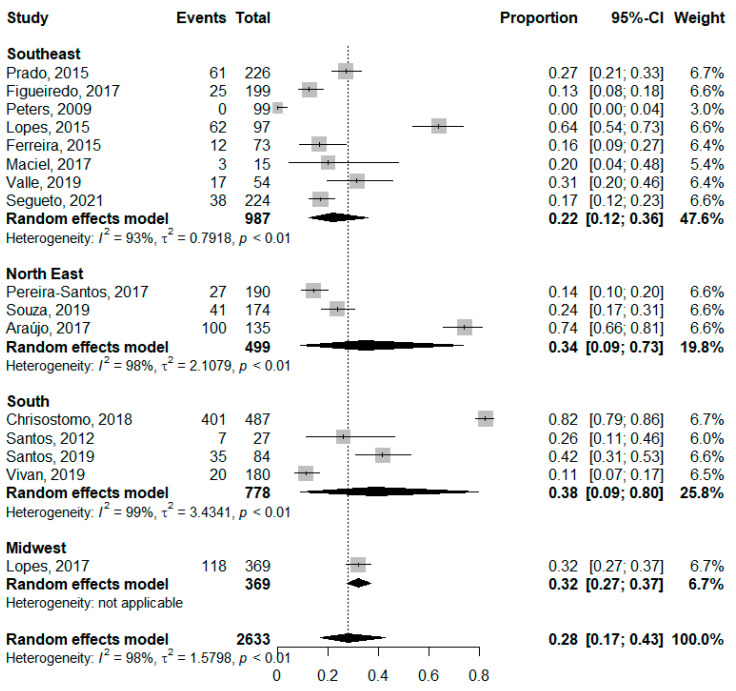
Prevalence of Vitamin D deficiency in childbearing women stratified by Brazilian geographic region. [33,34,35,36,37,39,40,41,42,43,44,45,46,47,48].

**Figure 4 nutrients-14-04351-f004:**
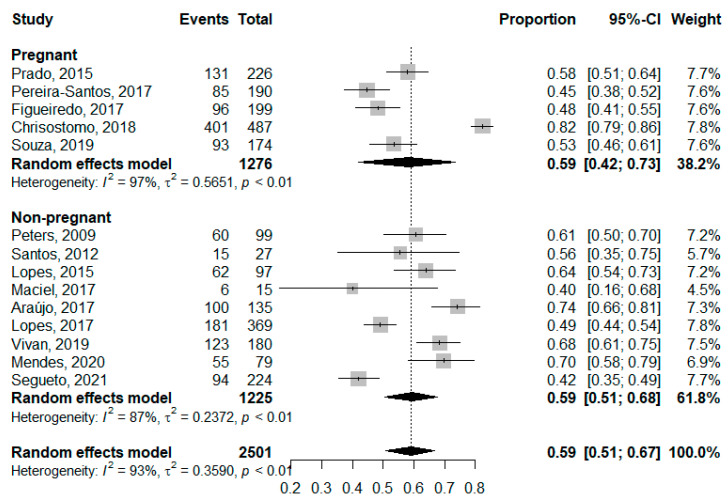
Prevalence of Vitamin D insufficiency in childbearing women stratified by pregnancy status. [33,34,35,36,39,40,41,43,44,46,47,48,49].

**Figure 5 nutrients-14-04351-f005:**
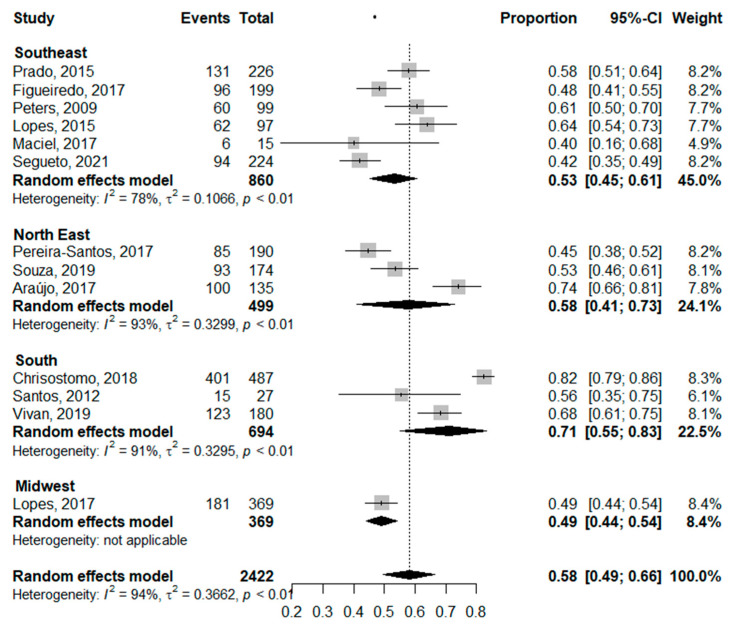
Prevalence of Vitamin D insufficiency in childbearing women stratified by Brazilian geographic region. [33,34,35,36,39,40,41,43,44,46,47,48].

**Figure 6 nutrients-14-04351-f006:**
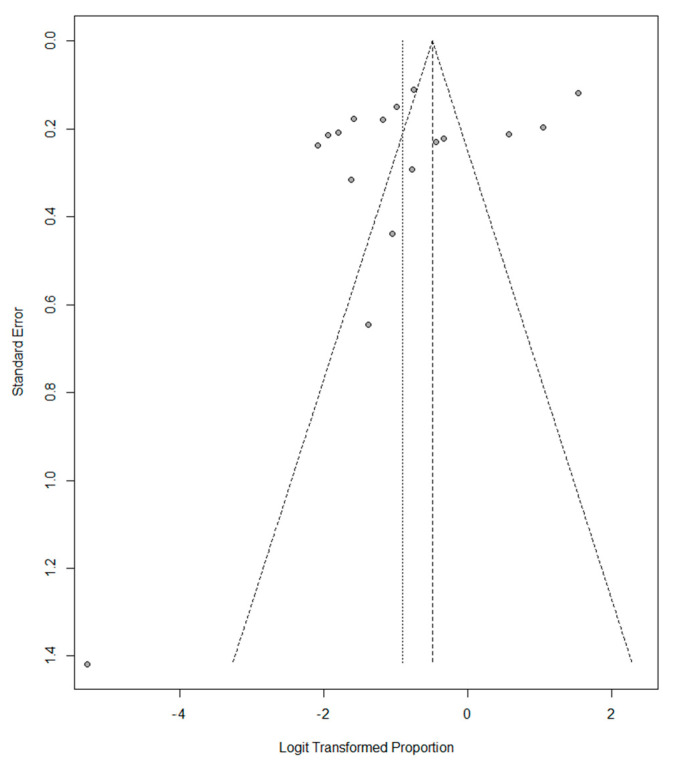
Funnel plot of Vitamin D deficiency in childbearing-aged women.

**Figure 7 nutrients-14-04351-f007:**
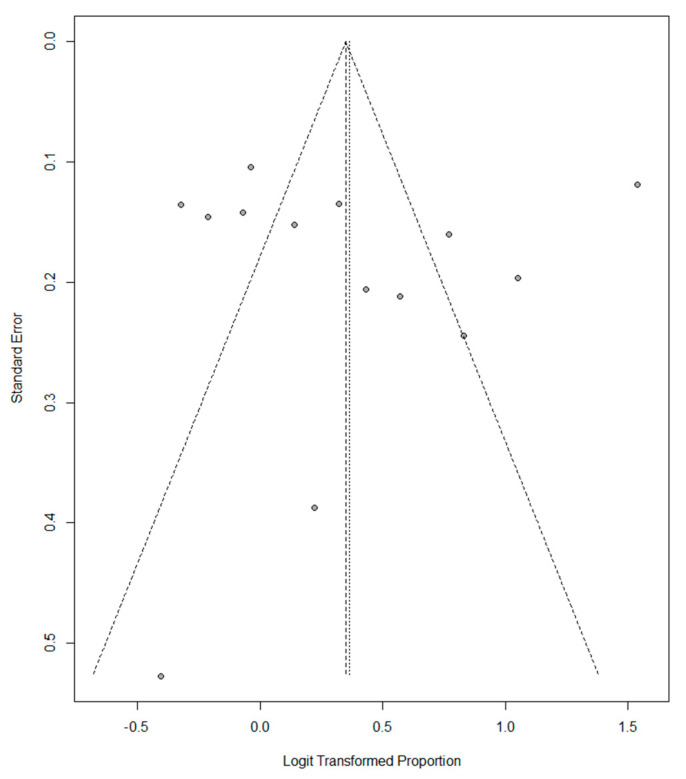
Funnel plot of Vitamin D insufficiency in childbearing-aged women.

**Table 1 nutrients-14-04351-t001:** Summarizing the association factors as risk or protective in pregnant and non-pregnant women.

Author/Year/Location	Investigated Variables	Summary of the Association of Vitamin D Deficiency or Insufficiency
	**Pregnant women**	
Pereira-Santos/2017 [35] San Antonio de Jesus-Bahia Northeast	Age, MFI, YS, skin color, MS, GA, number of weekly SEs, region of body exposed to sun, SY, means of transport	Association–risk factor to: Being married or in a relationship Face and hands on the sun Use of vehicle as transport Season-winter
Chrisostomo/2018 [33] Curitiba-Paraná South	Age, ethnic origin, skin phototype according to the Fitzpatrick classification, tobacco exposure, AI, YS, PCI, SY; clinical data: preeclampsia, DM, HIV, BMI, medication use, stage of pregnancy, parity, number of spontaneous abortions, NP	Association–risk factor to: Preeclampsia Serum vitamin D Increase when blood collection is in the summer
Souza/2019 [34] São Luís-Maranhão Northeast	Marital status, skin color, PCI, religion, sunscreen use, adolescence, NP, gestational trimester	Lower mean Vitamin D–risk factor to: Religion/protestant Primiparous Association hypovitaminosis-risk factor to: Adolescents Primiparous, Income
Prado/2015 [43] Viçosa-Minas Gerais Southeast	Women: age, skin color, place of residence, parity, supplementation, US, SE, education, MS, TD, AP, Ca, PTH, P	No association
Figueiredo/2017 [44] Rio de Janeiro Southeast	Age, skin color, YS, PCI, parity, smoking in the 1st trimester, AI in the 1st trimester, PA before pregnancy, SY, daily calcium and VD intake	No association Only increase of serum Vitamin D during the third trimester of pregnancy stated in winter or spring
	**Non-pregnant women**	
Ferreira/2015/Rio de Janeiro-RJ [45] Southeast	Age, skin color, AI, daily Ca consumption, creatinine, TPL, albumin, globulin, intracellular Ca, serum Ca, ionic Ca, urinary Ca/creatinine, PTH, W, BMI, BF, WC, HC, WC/HC ratio, WC/height ratio, glucose, insulin, HOMA-IR, TCL, HDL, LDL, TG, leptin, adiponectin, hs-CRP, RHI, SBP, DBP.	Association–risk factor to: Serum glucose, Homeostatic model assessment for insulin resistance
Araújo/2017 [36] João Pessoa-PB Northeast	Age, skin color, daily hours of sleep, SE, PA, daily VD intake, weight, height, BMI	Association–risk factor to: Calcium
Vivan **/2019 [46] Porto Alegre-RS South	Age, sex, skin color, educational level, SY, BMI, SAH, DM, multiple comorbidities, use of thiazide diuretics, use of Ca channel blocker, Hb1Ac, FBG	Association–risk factor to: Non-white skin Diabetes Use of calcium channel blocker Glycated hemoglobin Fasting blood glucose Autumn-winter Body mass index
Fonseca Valle/2019 [37] Rio de Janeiro-RJ Southeast	Age, weight, height, BMI, WC, waist-to-height ratio, CI, PTH, LDL, HDL, TCL, TG, SBP, DBS, FBG, Hb1Ac, SI, HOMA-IR,	Association–risk factor to: Weight Body mass index Waist circumference Waist-to-height ratio Systolic blood pressure Homeostatic model assessment for insulin resistance
Peters/2009 [39] Indaiatuba-SP Southeast	Sunscreen, physical exercise, sun exposure, DBS, SBP	No association
Santos **/2012 [40] Curitiba-PR, Porto Alegre-RS South	Age, height, BMI, age at menarche, thelarche, SY, genotype, WC	No association
Lopes/2017 [41] Brasilia DF Midwest	Infertility (low ovarian reserve, PCOS, tubal factors, endometriosis, multiple factors, unexplained infertility)	No association
Santos/2019 [42] Brazil South	BP, weight, height, WC, BMI, TCL, LDL, HDL, TG, HOMA-IR, estradiol, testosterone, SHBG, PTH, VDBP, albumin	No association
Segheto **/2021 [38] Viçosa-Minas Gerais Southeast	Bone mass, total bone mineral content	No association
Lopes/2015 [47] São Paulo-SP Southeast	Not investigated	
Maciel/2017 [48] Jacareí-SP Southeast	Not investigated	
Mendes/2020 [49] several cities in Brazil	Not investigated	

** Data obtained by contacting authors. Abbreviations—BMI: body mass index; BP: blood pressure; CI: conicity index; DBP: diastolic blood pressure; DEFN: deficiency; FBG: fasting blood glucose; Hb1Ac: glycated hemoglobin; HDL: high-density lipoprotein; HOMA-IR: homeostatic model assessment for insulin resistance; HVD: hypovitaminosis D; INSUF: insufficiency; LDL: low-density lipoprotein; PR: prevalence ratio; PTH: parathyroid hormone; SBP: systolic blood pressure; SHBG: sex hormone binding globulin; TCL: total cholesterol levels; TG: triglycerides; VD: vitamin D; VDBP: vitamin D binding protein; VDD: vitamin D deficiency; WC: waist circumference.

**Table 2 nutrients-14-04351-t002:** Score of studies according to quality criteria by Downs and Black and Grade methods.

Study (Year)	Conflict of Interests	Ethical Approval	Downs and Black Checklist																		GRADE
			A/1	B/2	C/3	D/5	E/6	F/7	G/9	H/10	I/11	J/12	K/17	L/18	M/20	N/21	O/25	P/26	Total	Score #	
Segheto et al. (2021) [39]	*	YES	1	1	1	2	1	1	-	1	1	1	-	1	1		1		13/13	100%	●●●
Mendes et al. (2020) [49]	*	YES	1	1	1	2	1	1	-	1	0	0	-	1	1	-	1	-	11/13	85%	●●●
De Souza; Silva; Figueiredo (2019) [44]	*	YES	1	1	1	0	1	1	-	1	0	0	-	1	1	-	0	-	8/13	62%	●●
Valle; Giannini (2019) [37]	*	NO	1	1	1	0	1	1	-	1	0	1	-	1	1	-	1	-	10/13	77%	●●
Vivan et al. (2019) [46]	*	YES	1	1	1	1	1	1	-	1	0	0	-	1	1	-	1	-	10/13	77%	●●●●
Christostomo et al. (2018) [33]	*	YES	1	1	1	1	1	1	-	1	0	1	-	1	1	-	1	-	11/13	85%	●●●
Figueiredo et al. (2017) [44]	*	YES	1	1	1	2	1	1	1	1	0	0	1	1	1	1	1	1	15/17	88%	●●●
Araújo et al. (2017) [36]	*	YES	1	1	1	1	1	1	-	1	0	0	-	1	1	-	1	-	10/13	77%	●●●
Pereira-Santos et al. (2017) [35]	*	YES	1	1	1	1	1	1	-	1	1	1	-	1	1	-	1	-	12/13	92%	●●●
Lopes et al. (2017) [41]	*	NO	1	1	1	1	1	1	-	1	0	1	-	1	1	-	1	-	11/13	85%	●●●
Maciel; Reis (2017) [48]	*	YES	1	1	1	0	1	1	-	0	0	0	-	1	1	-	0	-	7/13	54%	●
Ferreira et al. (2015) [45]	*	YES	1	1	1	2	1	1	-	1	0	0	-	1	1	-	1	-	11/13	85%	●●●
Prado et al. (2015) [43]	*	YES	1	1	1	2	1	0		1	1	1		1	1		1		12/13	92%	●●●
Lopes et al. (2015) [47]	*	YES	1	1	1	1	1	1	-	1	0	1	-	1	1	-	1	-	11/13	85%	●●●
Santos et al. (2012) [40]	*	YES	1	1	1	2	1	1	-	1	0	0	-	1	1	-	1	-	11/13	85%	●●●
Peters et al. (2009) [39]	*	YES	1	1	1	1	1	1		1	0	0		1	1		1		10/13	77%	●●

Downs and Black checklist: (A) objective clearly stated; (B) main results clearly described; (C) sample characteristics clearly defined; (D) distribution of the main confounders clearly described; (E) main findings clearly defined; (F) random variability in the estimates provided; (G) loss of follow-up described; (H) reported probability values; (I) representative target sample of the population; (J) recruitment of a representative sample of the population; (K) analyses adjusted for different durations of follow-up; (L) statistical tests used properly; (M) valid/reliable primary results; (N) sample recruited from the same population; (O) adequate adjustment for confusion; and (P) sample follow-up losses are taken into account (corresponding to items 1–3, 5–7, 9–12, 17,18, 20, 21, 25, and 26). Items G and P were applied only to cross-sectional studies. Items K and N were applied only to case–control and cross-sectional studies. # The score reaches 100% with 13, 15, and 17 points for cross-sectional, case–control, and cross-sectional studies, respectively. *, not reported, -, not applicable. GRADE, Grading of Recommendations, Assessment, Development, and Evaluations; a filled circle, very low quality; two filled circles, poor quality; three filled circles, moderate quality; four filled circles, high quality.

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
