# Peer review of "Prevalence of Vitamin D and Calcium Deficiency and Insufficiency in Women of Childbearing Age and Associated Risk Factors: A Systematic Review and Meta-Analysis"

_nutrients, 2022, doi:10.3390/nu14204351_

Round 1
Reviewer 1 Report
Da Silveira et al- Systematic review
Prevalence of vitamin D and calcium deficiency and insufficiency in women of childbearing age and associated risk factors: a systematic review and meta-analysis
Summary of paper
The authors aimed to estimate the prevalence of 25 serum vitamin D and calcium deficiencies and insufficiencies and associated risk factors in Brazilian women of childbearing age and to assess whether there are differences in prevalence according 27 to regions of the country and the presence or absence of pregnancy. The paper is a systematic review and meta-analysis, PRISMA guidelines are used, protocol is registered in Prospero.
The paper is highly relevant and the systematic review as well as meta-analysis are well conducted in terms of methods. I very much enjoyed reading it and I find the results interesting. In my opinion, however, some changes could be recommended before publication.
Major concerns
Tables S1 and S2 are very difficult to read, it is hard to get an overview of the associated variables. I suggest making two smaller, more compressed and organized versions, which ideally should fit in one A4 page if possible. An additional comparative table between pregnant and non-pregnant women would also be ideal.
Do the authors have any insights in Vitamin D dietary intake in the investigated population? What are the public health recommendations regarding Vitamin D and Calcium supplementation in Brazil, and what are the Vitamin D and Calcium contents in the normal Brazilian diet?
Minor concerns
Overall, the paper could benefit from proofreading by a native English speaker
Line 47, define “deficiency” (% with belox xx nmol/l)
Line 57, cite Andersen et al Vitamin D insufficiency is associated with increased risk of first-trimester miscarriage in the Odense Child Cohort | The American Journal of Clinical Nutrition | Oxford Academic (oup.com)
Line 137, in addition please cite Andersen et al Parity and tanned white skin as novel predictors of vitamin D status in early pregnancy: a population‐based cohort study - Andersen - 2013 - Clinical Endocrinology - Wiley Online Library
Line 72, ”Our review differs from the previous, seeing as it is…”
Author Response
Reviewer #1
Summary of paper
The authors aimed to estimate the prevalence of 25 serum vitamin D and calcium deficiencies and insufficiencies and associated risk factors in Brazilian women of childbearing age and to assess whether there are differences in prevalence according 27 to regions of the country and the presence or absence of pregnancy. The paper is a systematic review and meta-analysis, PRISMA guidelines are used, protocol is registered in Prospero. The paper is highly relevant and the systematic review as well as meta-analysis are well conducted in terms of methods. I very much enjoyed reading it and I find the results interesting. In my opinion, however, some changes could be recommended before publication.
Major concerns
Tables S1 and S2 are very difficult to read, it is hard to get an overview of the associated variables. I suggest making two smaller, more compressed, and organized versions, which ideally should fit in one A4 page if possible. An additional comparative table between pregnant and non-pregnant women would also be ideal.
Response: Thank you for this valuable suggestion. The authors have made a new Table 1 summarizing the evidence on the associated variables comparing pregnant and non-pregnant women on the same Table as shown below:
Table 1: Summary of the risk factors associated to vitamin D deficiency or insufficiency in pregnant and non-pregnant women
Author/Year/Location |
Investigated variables |
Summary of the association of vitamin D deficiency or insufficiency |
Pregnant women |
||
Pereira-Santos/2017[33] San Antonio de Jesus-Bahia
Northeast region |
Age, MFI, YS, skin color, MS, GA, number of weekly SEs, region of body exposed to sun, SY, means of transport |
Association – risk factor to: Being married or in a relationship Face and hands on the sun Use of vehicle as transport Season - winter |
Chrisostomo/2018 [35] Curitiba-Paraná
South region |
Age, ethnic origin, skin phototype according to the Fitzpatrick classification, tobacco exposure, AI, YS, PCI, SY; clinical data: preeclampsia, DM, HIV, BMI, medication use, stage of pregnancy, parity, number of spontaneous abortions, NP |
Association – risk factor to: Preeclampsia
Serum vitamin D Increase when blood collection is in the summer |
Souza/2019 [36] São Luís-Maranhão
Northeast region |
Marital status, skin color, PCI, religion, sunscreen use, adolescence, NP, gestational trimester |
Association lower mean Vitamin D– risk factor to: Religion/protestant Primiparous Association hypovitaminosis - risk factor to: Adolescents Primiparous, Income |
Prado/2015 [32] Viçosa-Minas Gerais
Southeast region |
Women: age, skin color, place of residence, parity, supplementation, US, SE, education, MS, TD, AP, Ca, PTH, P |
No association |
Figueiredo/2017[34] Rio de Janeiro
Southeast region |
Age, skin color, YS, PCI, parity, smoking in the 1st trimester, AI in the 1st trimester, PA before pregnancy, SY, daily calcium and VD intake |
No association Only increase of serum Vitamin D during the third trimester of pregnancy stated in winter or spring |
|
Non-pregnant women |
|
Ferreira/2015/Rio de Janeiro-RJ [32]
Southeast region |
Age, skin color, AI, daily Ca consumption, creatinine, TPL, albumin, globulin, intracellular Ca, serum Ca, ionic Ca, urinary Ca/creatinine, PTH, W, BMI, BF, WC, HC, WC/HC ratio, WC/height ratio, glucose, insulin, HOMA-IR, TCL, HDL, LDL, TG, leptin, adiponectin, hs-CRP, RHI, SBP, DBP. |
Association – risk factor to: Serum glucose, Homeostatic model assessment for insulin resistance |
Araújo/2017 [42] João Pessoa-PB
Northeast region |
Age, skin color, daily hours of sleep, SE, PA, daily VD intake, weight, height, BMI |
Association – risk factor to: Calcium |
Vivan**/2019 [45] Porto Alegre-RS
South region |
Age, sex, skin color, educational level, SY, BMI, SAH, DM, multiple comorbidities, use of thiazide diuretics, use of Ca channel blocker, Hb1Ac, FBG |
Association – risk factor to: Non-white skin Diabetes Use of calcium channel blocker Glycated hemoglobin Fasting blood glucose Autumn-winter Body mass index |
Fonseca Valle/2019 [46] Rio de Janeiro-RJ
Southeast region |
Age, weight, height, BMI, WC, waist-to-height ratio, CI, PTH, LDL, HDL, TCL, TG, SBP, DBS, FBG, Hb1Ac, SI, HOMA-IR, |
Association – risk factor to: Weight Body mass index Waist circumference Waist-to-height ratio Systolic blood pressure Homeostatic model assessment for insulin resistance |
Peters/2009 [37] Indaiatuba-SP
Southeast region |
Sunscreen, physical exercise, sun exposure, DBS, SBP |
No association |
Santos**/2012 [38] Curitiba-PR, Porto Alegre-RS
South region |
Age, height, BMI, age at menarche, thelarche, SY, genotype, WC |
No association |
Lopes/2017 [43] Brasilia DF
Midwest region |
Infertility (low ovarian reserve, PCOS, tubal factors, endometriosis, multiple factors, unexplained infertility) |
No association |
Santos/2019 [44] Brazil
South region |
BP, weight, height, WC, BMI, TCL, LDL, HDL, TG, HOMA-IR, estradiol, testosterone, SHBG, PTH, VDBP, albumin |
No association |
Segheto**/2021 [48] Viçosa-Minas Gerais
Southeast region |
Bone mass, total bone mineral content |
No association |
Lopes/2015 [39] São Paulo-SP
Southeast region |
Not investigated |
|
Maciel/2017 [41] Jacareí-SP
Southeast region |
Not investigated |
|
Mendes/2020 [47] several cities in Brazil |
Not investigated |
|
*Population includes sample size, age group, and sample origin. Abbreviations- AI: alcohol intake); BMI: body mass index; BP: blood pressure; CI: conicity index; DBP: diastolic blood pressure; DEFN: Deficiency; FBG: fasting blood glucose; Hb1Ac: glycated hemoglobin; HDL: high-density lipoprotein; HOMA-IR: homeostatic model assessment for insulin resistance; HVD: hypovitaminosis D; INSUF: insufficiency; LDL: low-density lipoprotein; PCI: per capita income; PR: prevalence ratio; PTH: parathyroid hormone; SBP: systolic blood pressure; SHBG: sex hormone binding globulin; SY: season of the year; TCL: total cholesterol levels; TG: triglycerides; VD: vitamin D; VDBP: vitamin D binding protein; VDD: vitamin D deficiency; WC: waist circumference; YS: years of study.
The authors kept as a supplementary file those Tables with all the detailed information as they can be useful to other researchers and/or to update this systematic review in the future.
Do the authors have any insights in Vitamin D dietary intake in the investigated population? What are the public health recommendations regarding Vitamin D and Calcium supplementation in Brazil, and what are the Vitamin D and Calcium contents in the normal Brazilian diet?
Response: Thank you. We would like to emphasize that the focus of our systematic review was to assess serum vitamin D deficiency and insufficiency. However, to address your comment on food intake as one of the factors that influence vitamin D deficiency, we have included the following sentences in the revised Discussion:
“Another influencing factor is vitamin D dietary intake. The average consumption of this nutrient in adult Brazilians ranged from 2.4 to 4.67µg [34,37,47], which is below the recommended value of Estimated Average Requirement of 10 µg/day for women of childbearing age (15-49 years), and also according to Dietary Reference Intakes (DRIs) of 15µg/day for women [56]. A study that evaluated Latin American women pointed to an average vitamin D intake of 1.9 µg in Brazil [57], and in another that evaluated dietary intake in Brazilian adolescents (15-19 years old), observed a median intake of 1.48 µg/day of vitamin D [36]. However, dietary intake was not associated with vitamin D deficiency or insufficiency [34,70].” (Line 356 to 364)
“The public health program in Brazil on vitamin D food fortification or supplementation is not mandatory [34,39] and not usual among women receiving prenatal care in public service [34]. Furthermore, the implementation of such program is challenging due to differences in vitamin D serum levels recommendations which are based on studies conducted in high latitude countries with older Caucasian populations [47]. Therefore, further studies are needed concerning the optimal cut-offs for specific vitamin D levels for the Brazilian population [47].” (Lines 390 to 396)
“Mean dietary calcium intake was 625.1 to 738.72 mg/day [37,40,47]. According to Herrera-Cuenca (2021), 95.16% of Brazilian women had an inadequate calcium intake, which was also below the current recommendation (1,000-1,200 mg/day).” (Line 378 to 381)
Minor concerns
Overall, the paper could benefit from proofreading by a native English speaker
Response: Thank you. The manuscript has been revised by a native English speaker as requested.
Line 47, define “deficiency” (% with below xx nmol/l)
Response: Thank you. We have included the deficiency definition suggested as follows: “Although there is no agreement on the optimal range of vitamin D deficiency, it is predominantly characterized by serum 25(OH)D concentrations below 25–30 nmol /L (10–12 ng/mL) [7]”.(Line 47 to 49)
Line 57, cite Andersen et al Vitamin D insufficiency is associated with increased risk of first-trimester miscarriage in the Odense Child Cohort | The American Journal of Clinical Nutrition | Oxford Academic (oup.com)
Response: Thank you for suggesting this reference. We have included in the revised manuscript the negative outcome of vitamin D insufficiency based on the suggested reference: “such as first trimester miscarriages” (Lines 54 and 55)
Line 72, “Our review differs from the previous, seeing as it is…”
Response: Thank you. We have included in the revised manuscript the following sentence:
“Our review differs from the previous, as it included rigorous data from well-conducted studies stratified by gestation status and presented regional differences by the five great geographic Brazilian regions”. (Line 77 to 80)
Reviewer 2 Report
This is a review that try to explore the prevalence of vitamin D deficiency and insufficience in women of childbearing age in a specific country and possible risk factor associated. The review is interesting, but there are different major critical issued to be addressed that make not suitable the paper in present form for publication in journal
1. It could be useful a revision of the English language.
2. The introduction is lack of information about the role of vitamin D. The clinical implication in different muscoloskeletal diseases and birth has been demonstrated in different recent studies. I suggest to deep this topic with a more robust literature.
2. Please, add the used mash terms for the research
3. The authors should be discuss the possible bias of the supplementation of vitamin D in each population. This point should be addressed
4. It should be interesting to report in table the exact region of each included article
4. The metanalysis is quite confuse. The authors want to demnostrate the difference in vitamin D status in different region and in different women status (pregnant-no pregnant), but the affirm in the PICO model that the comparator was pregnant-not pregnant, without mention in region statusr
5. The aim of the study should be the evalutation of serum level of Vitamin D and associated risk factor. Why have you included studies that not included this evalutation. Moreover, the authors want to lead a systemati review and metanalysis on the prevalence of vitamin D/serum calcium and risk factor associated in childbearing age, but the metanalysis is not clear in this term and is not the right form for the aim
6. The discussion should address the clinical implication of your research
Best regards
Author Response
This is a review that try to explore the prevalence of vitamin D deficiency and insufficiency in women of childbearing age in a specific country and possible risk factor associated. The review is interesting, but there are different major critical issued to be addressed that make not suitable the paper in present form for publication in journal
- It could be useful a revision of the English language.
Response: Thank you. The paper has been revised by a native English speaker as suggested.
- The introduction is lack of information about the role of vitamin D. The clinical implication in different musculoskeletal diseases and birth has been demonstrated in different recent studies. I suggest to deep this topic with a more robust literature. Please, add the used mash terms for the research
Response: Thank you. We agree with the importance of discussing further the topic mentioned by the reviewer. Therefore, we have rewritten part of the introduction as follows:
“The evidence showing that vitamin D deficiency can trigger negative outcomes for both the mother and child, including hypertension in women of childbearing age [9], gestational diabetes [10] and negative adverse perinatal outcomes [11] such as first trimester miscarriages [12] has resulted in a growing public health concern at primary care level and the need of public policies at the national level. Vitamin D deficiency has also been associated with some extra-skeletal damages, such as osteoporosis, chronic musculoskeletal pain, muscle weakness, and an increased risk of falling [13-16]. The negative adverse perinatal outcomes [17] include impairment of anthropometric measurements of the neonate such as birth weight, length, and head circumference [18] in addition to a higher risk of premature rupture of membranes [10] and miscarriage [19]. At birth, it can be related to infantile eczema, nutritional rickets, severe hypocalcemia, and other orthopedic complications in neonates and children [20].” (Line 52 to 63)
- The authors should be discussing the possible bias of the supplementation of vitamin D in each population. This point should be addressed
Response: Thank you. The authors agree with the point raised by the reviewer. Therefore, we have included the following sentence in the revised Discussion:
“In most of the studies included in this systematic review, women who reported vitamin D supplementation were excluded from the study samples [34,35,37,39,40,42,43,45-47]. Therefore, the lack of data on vitamin D dietary intake and supplementation may be a potential source of bias, as it may interfere with vitamin D metabolism [65-67].” (Lines 411 to 417)
- It should be interesting to report in table the exact region of each included article
Response: Thank you for your suggestion. We have included the exact geographic region in the revise Table 1.
- The metanalysis is quite confuse. The authors want to demonstrate the difference in vitamin D status in different region and in different women status (pregnant-no pregnant), but the affirm in the PICO model that the comparator was pregnant-not pregnant, without mention in region status.
Response: Thank you for raising this important point. The PICO model was reformulated to clarify that our objective was to assess the prevalence of vitamin D and calcium deficiency by geographic region and pregnancy status i.e. pregnant or non-pregnant.
“PICO structure, with “P” (population) being women of childbearing age, “I” being no intervention, “C” being comparison between pregnant and non-pregnant subgroups and geographic region, and “O” (outcome) prevalence calcium and vitamin D deficiency.” (Lines 93 to 96)
- The aim of the study should be the evaluation of serum level of Vitamin D and associated risk factor. Why have you included studies that not included this evaluation? Moreover, the authors want to lead a systematic review and metanalysis on the prevalence of vitamin D/serum calcium and risk factor associated in childbearing age, but the metanalysis is not clear in this term and is not the right form for the aim
Response: Thank you for the opportunity to clarify this point. The aim of this study was to evaluate the prevalence of serum vitamin D and calcium levels in pregnant and non-pregnant women according to the 5 geographic regions of Brazil. We also intended to identify the factors associated with vitamin D and calcium deficiency. However, most of the studies had information on the prevalence but not on the associated factors. Therefore, we do not have enough data to perform a meta-analysis of the prevalence of calcium and the factors associated with vitamin D and calcium deficiency. We have included this information in the revised Discussion as a limitation.
“Furthermore, it was not possible to perform a meta-analysis of calcium deficiency and the factors associated with vitamin D and calcium deficiency in women of childbearing age due to the low number of studies found.” (Lines 415 to 417)
- The discussion should address the clinical implication of your research
Response: Thank you for your careful review. The authors have included the following sentences in the revised Discussion to address your suggestion:
“This study highlights the high prevalence of vitamin D insufficiency in both pregnant and non-pregnant women. Therefore, there is a clear need for the development of public policies, with a focus on preventing and minimizing this public health concern. From the clinical implication point of view, health education campaigns at national level, focusing on lifestyle changes such as 10 minutes of daily sun exposure and a balanced diet [68,69], outdoor physical activity practices [22], may be sufficient to increase knowledge among women and promote adequate serum levels of vitamin D and calcium.” (Line 429 to 435)
“Education and health promotion, including dietary intake that meets nutrient and micronutrient needs, with the use of supplementation and food fortification, if necessary, may result in improvements of nutritional deficiencies in general [25,57, 70]. However, the level of knowledge about calcium and vitamin D deficiency and associated factors in Brazilian women varies considerably, making it harder to promote public policies, particularly among different ethnic/racial groups due to insufficient evidence [34,47]. In this sense, the results of this study may contribute to fostering policies to be increased, aiming at the development of research on calcium and vitamin D deficiency in Brazilian women of childbearing age, as well as for the identification of potential risk and protection factors. It is important to mention that no research was carried out in the North region and only two in the Midwest region. Therefore, there is a clear need for research on serum calcium deficiency in all Brazilian regions and vitamin D deficiency in the North, Northeast and Midwest regions to provide reliable data to plan public health actions.” (436 to 448)
“According to the Brazilian Society of Endocrinology and Metabolism, the diagnosis of serum vitamin D deficiency should be performed in pregnant and lactating women [23]. Considering that the prevalence of vitamin D deficiency found in this systematic review did not differ between pregnant and non-pregnant women, it is necessary that women of reproductive age are also tested to monitor their vitamin D nutritional status.” (Line 449 to 452)

Round 2
Reviewer 2 Report
Dear Authors,
after a carefully lecture of your revised paper, I was impressed of your response and of the modification of the paper that have raised the scientific soundness. At the light of my knowledge, I suggest only minor revision in introduction section:
Introduction: I suggest to spend few lines in the role of vitamin D in women and its link with oral health. Please read "1) Ferrillo M, Migliario M, Roccuzzo A, Molinero-Mourelle P, Falcicchio G, Umano GR, Pezzotti F, Foglio Bonda PL, Calafiore D, de Sire A. Periodontal Disease and Vitamin D Deficiency in Pregnant Women: Which Correlation with Preterm and Low-Weight Birth? J Clin Med. 2021 Oct 2;10(19):4578. doi:10.3390/jcm10194578, "Ferrillo M, Migliario M, Marotta N, Lippi L, Antonelli A, Calafiore D, Ammendolia V, Fortunato L, Renò F, Giudice A, Invernizzi M, de Sire A. OralHealth in Breast Cancer Women with Vitamin D Deficiency: A Machine Learning Study. J. Clin. Med. 2022; 11(16):4662. doi: 10.3390/jcm11164662", and "de Sire A, Gallelli L, Marotta N, Lippi L, Fusco N, Calafiore D, Cione E, Muraca L, Maconi A, De Sarro G, Ammendolia A, Invernizzi M. Vitamin D Deficiency in Women with Breast Cancer: A Correlation with Osteoporosis? A Machine Learning Approach with Multiple Factor Analysis. Nutrients. 2022 Apr 11;14(8):1586. doi: 10.3390/nu14081586.
Methods: thank you for your revision
Results: Thank you for your revision
Discussion: well done
Best Regards
Author Response
Reviewer 2
Introduction: I suggest to spend few lines in the role of vitamin D in women and its link with oral health. Please read "1) Ferrillo M, Migliario M, Roccuzzo A, Molinero-Mourelle P, Falcicchio G, Umano GR, Pezzotti F, Foglio Bonda PL, Calafiore D, de Sire A. Periodontal Disease and Vitamin D Deficiency in Pregnant Women: Which Correlation with Preterm and Low-Weight Birth? J Clin Med. 2021 Oct 2;10(19):4578. doi:10.3390/jcm10194578.
Response: Thanks for your suggestion. We have included the following sentence in the introduction:
“Therefore, vitamin D deficiency in combination with periodontal disease has been associated with preterm and low birth weight [21]”(Lines 64 and 65).